# Process-level model evaluation: A Snow and Heat Transfer Metric

Andrew G. Slater[1], David M. Lawrence[2], Charles D. Koven[3]

[1] NSIDC/CIRES, University of Colorado, Boulder, 80303, USA
[2] National Center for Atmospheric Research, Boulder, 80305, USA
[3] Lawrence Berkeley National Laboratory, Berkeley, 94720, USA

*Correspondence to*: David Lawrence (dlawren@ucar.edu)

**Abstract.** Land models require evaluation in order to understand results and guide future development. Examining functional relationships between model variables can provide insight into the ability of models to capture fundamental processes and aid in minimizing uncertainties or deficiencies in model forcing. This study quantifies the proficiency of land models to appropriately transfer heat from the soil through a snowpack to the atmosphere during the cooling season (Northern Hemisphere: October-March). Using the basic physics of heat diffusion, we investigate the relationship between seasonal amplitudes of soil versus air temperatures due to insulation from seasonal snow. Observations demonstrate the anticipated exponential relationship of attenuated soil temperature amplitude with increasing snow depth and indicate that the marginal influence of snow insulation diminishes beyond an 'effective snow depth' of about 50cm. A Snow and Heat Transfer Metric (SHTM) is developed to quantify model skill compared to observations. Land models within the CMIP5 experiment vary widely in SHTM scores and deficiencies can often be traced to model structural weaknesses. The SHTM value for individual models is stable over 150 years of climate 1850-2005, indicating that the metric is insensitive to climate forcing and can be used to evaluate each model's representation of the insulation process.

## 1 Introduction

The current generation of land models are typically complex in nature and simulate a vast array of processes [Clark et al., 2015; Prentice et al., 2015]. Interdependencies within these models produce external and internal feedbacks that can operate on various temporal and spatial scales. It is therefore imperative that such models be rigorously evaluated in order to interpret their performance, as well as to guide future development.

Verifying a model result against observations using statistics such as root mean squared error can provide useful information, but alone does not necessarily indicate whether the model is getting the right answers for the right reason [Abramowitz et al., 2008; Gupta et al., 2008]. For example, the impact of the forcing data can play an equal or greater role in results than some aspects of the model [Ménard et al., 2015]. Most global-scale "observationally based" data sets contain substantial uncertainty, especially in the relatively sparsely observed high latitudes [Anisimov et al., 2007; Decker et al., 2012; Slater,

2016; Mudryk et al. 2015], and coupled Earth System Models can produce biased surface meteorology, particularly in high latitude winters [Slater and Lawrence, 2013]. It therefore follows, that if one is trying to assess land model structural error (i.e., deficiencies in model architecture and/or parameterizations) it is preferable to reduce ambiguity by minimizing other sources of uncertainty, e.g. uncertainties in model parameters, initial conditions or forcing (in either land-only or coupled simulations).

In high latitude regions an important process is snow and soil heat transfer. The temperature and state of the soil, whether frozen or thawed, plays a pivotal role in the timing and magnitude of energy, mass and biogeochemical fluxes between the land and atmosphere [Hobbie et al., 2000; Dutta et al., 2006; Monson et al., 2006; Lawrence et al., 2012, 2015]. Concerns about the potential release of vast carbon stores currently locked in permafrost soils and its feedback on the climate has brought this topic to the fore [Schuur et al., 2015]. Model estimates of Arctic carbon fluxes are highly variable [Fisher et al., 2014; Koven et al., 2015] in part due to differences in simulation of soil temperatures and permafrost conditions. Essentially, we need to ensure soil temperature and moisture are correctly simulated before confidence in projected biogeochemical fluxes can be achieved. Additionally, biases in the simulated temperature at the snow/soil interface can adversely affect the snow pack itself though the impact of these biases on snow metamorphism at the base of the snow pack.

In cold mid-latitude and Arctic regions, snow forms an insulating barrier between the colder atmosphere and underlying soil during winter. The impact of snow on soil temperatures, particularly in permafrost regions, has been well documented in both observational [Sharratt et al., 1992; Zhang, 2005] and modeling studies [Luetschg et al., 2008; Lawrence and Slater, 2010; Ekici et al., 2015; Yi et al., 2015]. A variety of structures has been used to represent snow in models [Slater et al., 2001], with varying levels of simulated insulation [Koven et al., 2013]. The aim of this work is to define a metric that demonstrates processes-level model evaluation using the heat transfer mechanism from atmosphere to soil under the conditions of a seasonal snowpack.

## 2 Method

The theory of conductive heat flow under periodic forcing (e.g., the annual cycle) is demonstrated in many texts [Lunardini, 1981; Yershov, 1998].Taking an example of a semi-infinite medium forced at the surface with a periodic temperature wave with no phase change or mass exchange, we can compute the amplitude of this periodic wave at soil depth ($z$) as:

$$A_z = A_o \, e^{-\left(\frac{z}{d}\right)} \tag{1}$$

where $A_z$ is the amplitude of the temperature wave at depth ($z$), $A_o$ is the surface amplitude and $d$ is the damping depth defined as the depth at which the $1/e$ of the surface amplitude occurs. The value of $d$ is a function of the thermal diffusivity of the medium and the length of time that the forcing is applied.

The above theory is adapted to the cooling period of the year, defined here as October to March. A vast portion of the terrestrial Northern Hemisphere becomes snow covered during October and remains so beyond March. The actual land surface temperature is rarely observed *in situ*, but 2m air temperature serves as a sufficient proxy as the two quantities tend to equilibrate towards each other, particularly in colder months of high latitude regions with low solar input, though in situations with strong inversions, the temperature difference between the snow-air interface and 2m height can be substantial
(this is an acknowledged, yet unavoidable limitation). Observations (Section 3) show that the coldest air temperature is typically in January or February, while the coldest soil temperature at 20cm depth usually lags by a month. Confining data to the annual cooling period, the amplitude of air temperature is taken as

$$A_{air} = MAX(T_{air}) - MIN(T_{air}) \tag{2}$$

The soil temperature amplitude ($A_{soil}$) can be calculated annually in the same way. To elucidate the process of heat transfer and remove climatically driven factors, for example the large seasonal cycle associated with deep continental regions as compared to more moderate coastal locations, a normalized temperature amplitude difference is computed as

$$A_{norm} = \frac{A_{air} - A_{soil}}{A_{air}} \tag{3}$$

which ranges from 0 to 1. $A_{norm}$ values near 0.0 indicate minimal difference in the annual cycle of air and soil temperatures, while a value close to 1.0 suggests soil temperatures essentially do not change over the cooling period. If we take $A_{soil}$ to be
$A_z$ and $A_{air}$ as $A_o$, then substitute equation (3) into equation (1) we arrive at the form where:

$$A_{norm} = 1. - e^{-\left(\frac{z}{d}\right)} \tag{4}$$

Such theory pertains to an idealized case, but in reality the distance $z$ would be affected by the quantity and temporal
sequence of snow accumulation and the damping depth $d$ will be impacted by snow density, soil inhomogeneity or phase change. Therefore, we propose a similar but more flexible approach:

$$A_{norm} = P + Q\left(1. - e^{-\left(\frac{S_{depth,eff}}{R}\right)}\right)$$ (5)

We introduce an offset, $P$, because even if there was absolutely no snow, the thermal properties of the soil, along with phase change, and imperfect heat transfer between the surface and near-surface air mean that the amplitude of soil temperature at a
depth (e.g., 20cm) will likely be different to that of the air. A multiplier, $Q$, is applied to account for the temporal nature of snow accumulation. Our analysis pertains to locations with seasonal, rather than permanent, snow cover so even if 3m of snow accumulates by the end of March there is likely to be some cooling in soil temperature from atmospheric forcing in October. The $R$ parameter is an effective damping depth of the snow and soil system. The density and morphology of the snow, along with soil moisture content and phase change play a role in determining $R$. The $R$ value also tends to govern the
marginal influence of additional snow insulation. To account for seasonal variations in the linear distance between the soil and the atmosphere (i.e., the snow depth $z$, in equation 4), we define an effective snow depth ($S_{depth,eff}$).

$S_{depth,eff}$ describes the insulation impact of snow and is an integral value such that the mean snow depth ($S$) each month ($m; 1$-
$6$) is weighted by its duration. The maximum duration ($M=6$) is the total cooling period of 6 months (Oct-Mar). The first
snow depth value ($S_1$) is the October mean snow depth.

$$S_{depth,eff} = \frac{\sum_{m=1}^{M}\left(S_m * (M + 1 - m)\right)}{\sum_{m=1}^{M} m}$$

(6)

As shown in Figure 1, a season with an early snowfall will typically produce a higher effective snow depth compared to a
linearly increasing snowpack with the same mean value. Similarly, if shallow snow persists for most of the winter but a large snowfall occurs in February, the effective snow depth will be lower than the linear case.

Given inputs of $A_{norm}$ and $S_{depth,eff}$ (from observations or models) we can efficiently compute the values of the three parameters $P$, $Q$, $R$ using a non-linear fitting method (e.g. the Levenberg-Marquardt (LM) algorithm [Press et al., 2007]).
The relationship between $A_{norm}$ and $S_{depth,eff}$ informs us about the heat transfer mechanism between the atmosphere and the subsurface. To alleviate the problem of overfitting, the curve is fit using data that is sampled evenly across a range of effective snow depths with up to 35 data points per 5cm of $S_{depth,eff}$ up to 50cm (with the final category being 45cm or beyond). This sampling was performed 100 times, with the median of fit values taken; if the LM algorithm does not converge for a given sample, another sample is taken. Sampling also ensures that a wide variety of climatic regimes is used
for characterizing the functional relationship seen in observations or a model.

## 3 Data

Assessing the relation between $A_{norm}$ and $S_{depth,eff}$ requires three pieces of data: screen level air temperature, 20cm soil temperature (as this is the most commonly observed depth) and snow depth. An analysis constraint is the need to use monthly mean values, as this is the most common output from large-scale land models.

A large network of hydrometeorological stations in Russia (and the Former Soviet Union) provide the three required variables as follows: air temperature was acquired from the NCDC Global Summary of the Day (https://data.noaa.gov/dataset/global-surface-summary-of-the-day-gsod), provided the snow depths measured at stations (as opposed to local transects) and information on soil temperatures is available in Sherstyukov and Sherstyukov, [2015]. Data

10 in the USA is from the Natural Resources Conservation Service as part of the SNOTEL and SCAN networks (http://www.wcc.nrcs.usda.gov/). In Canada, Alberta's AgroClimate Information Service (ACIS; http://agriculture.alberta.ca/acis/) collates and distributes station data within the province.

For a given season, we use only sites with complete, quality controlled records from October to March; 2049 observed site-

15 years are available. It is also recognized that the data has inherent limitations, for example it is unknown whether snow depths are measured at precisely the same location as the soil temperature measurements. Many measurements have been made on soils that have been historically altered, however our use of shallow (20cm) soil temperatures and focus on the winter period aims to minimizes possible artifacts due to disturbance.

Snow heat transfer in 13 models that participated in the CMIP5 experiment [Taylor et al., 2012; http://cmip-pcmdi.llnl.gov/cmip5/] is evaluated using the Historical (1850-2005) simulations. Soil columns within the various land models have different depths and layer thicknesses [Slater and Lawrence, 2013; their Figure 2] so a spline interpolation of each monthly mean temperature profile was used to estimate a 20cm value. Further details of the land models are available in Koven et al., [2013] and Slater and Lawrence, [2013].

All data is restricted to those instances where, during the cooling period (Oct-Mar), mean $T_{air}$ is below -1°C, mean $T_{soil}$ is below 2.5°C, $A_{air}$ is greater than 10°C and $S_{depth,eff}$ is greater than 1cm and less than 150cm. Observed locations fitting the above data criteria are shown in Figure 2. Model data meet the same criteria, but our intention is to test the ubiquitous physics of a process, thus model data is not restricted to the same locations as the observations and an order of magnitude

more (or greater) grid cells from the models are available for sampling.

## 4 Results and Discussion

The observations show the expected exponential shape and fit the underlying theory (5) well despite significant scatter in the data (Figure 3). The scatter likely arises from several sources including (1) the range of climate conditions and snow regimes [Sturm et al. 1995] that occur across the landscape, including the timing of snowfall and the pattern of snow metamorphism, (2) the properties and moisture content of the soil, and (3) uncertainties in the measurements themselves and the measurement locations of the observed data. It is not possible to distinguish which of these sources of uncertainty dominates the scatter seen in Figure 3 and it remains possible that snowpack dynamics in regions outside the sampled data would generate slightly different relationships, though the shape of the curve would not likely change. Grey shading in Figure 3 shows the span of median scatter of all 100 fits to the observations with the black curve being the median fit value.

The observational results demonstrate that the *marginal* influence of snow insulation relative to the annual cycle of air temperature diminishes beyond a $S_{depth,eff}$ of about 25cm. This phenomenon of insulation saturation has been previously noted in observations and models [Zhang, 2005; Lawrence and Slater, 2010].

Results from models are calculated the same as observations, with the median curve for each model shown on the same plot (Figure 4). The difference amongst model curves is considerable. Several models produce an exponential-like curve (e.g., CCSM4, GISS, MRI, NorESM), suggesting that they generally reflect the character of the observed relationship. Many models, however, do not reproduce the observed relationship. Models such as CanESM, GFDL, Hadley models, MIROC and MPI are more linear in their form. This group of poorer performing models all show $A_{norm}$ values of less than 0.45 at $S_{depth,eff}$ of 20cm (compared to a median observed value of 0.65 at 20cm depth). A low $A_{norm}$ value is produced when soil temperatures more closely track the changes in air temperature rather than being modulated by snow cover.

Note that the impact of heat transfer from the soil surface to 20cm depth can be inferred from the $A_{norm}$ values at $S_{depth,eff} =$ 0cm ($A_{norm}$ typically between 0.05 and 0.30 at $S_{depth,eff} =$ 0cm). Many models exhibit a lower normalized temperature difference value at $S_{depth,eff} =$ 0cm, which suggests that these models are transferring heat through the upper soil too efficiently. The potential sources of a soil heat transfer bias are myriad and could be due to biases/errors in soil texture, soil moisture, soil water phase, soil thickness, and vegetation. Most of these models, apart from CLM [Lawrence and Slater, 2008], do not represent the highly insulative soil organic matter, so this is a potential explanation of the common biases at $S_{depth,eff} =$ 0cm.

Incorrect snow heat transfer curves are symptomatic of model deficiencies. As an example, the land scheme in the Hadley Center models used here [MOSES2.2; Essery et al., 2001] applies a composite snow model where the top soil layer and snowpack share the same temperature [Slater et al., 2001], hence insulation is not properly accounted for and cold

temperatures easily penetrate into the soil. Conversely, the better performing models feature multi-layer snow packs that are more apt at emulating the nonlinear temperature profile of the snow pack. Note that the Hadley Centre model developers have addressed this limitation by implementing a multi-layer snow model [Best et al., 2011; Chadburn et al. 2015].

Despite the application of different coupled model component (e.g., atmosphere or ocean models) and initial conditions, both of which will influence terrestrial surface climate, the two Hadley Center climate models (HadGEM2-CC, HadGEM2-ES, both utilizing MOSES as their land model) produce essentially the same snow insulation curve. Similarly, CCSM4 and NorESM reproduce essentially the same curve, which is expected since both models utilize CLM4 as their land model. The fact that snow insulation curves are essentially the same for a particular land model, even when driven with different climatic

forcings from their parent climate model, suggests that the form of the curve does indeed capture the functional land model behavior.

The steep gradient of $A_{norm}$ at shallower $S_{depth,eff}$ is an important feature to capture as over 85% of seasonally snow covered land is estimated to have a $S_{depth,eff}$ less than 30cm. Reliable observed data of hemispheric scale snow depths or mass is not

yet available, but the prevalence of shallow snow can be seen in estimated climatological $S_{depth,eff}$ derived from numerous reanalyses (ERA-Interim [Dee et al., 2011], MERRA [Rienecker et al., 2011], CFSR [Saha et al., 2010], JRA-55 [Kobayashi et al., 2015], some of which assimilate snow depths), products such as GlobSnow [Takala et al., 2011] and station interpolations [Foster and Davy, 1988; Brown and Brasnett, 2010] (Figure 5).

**4.1 A Model Diagnostic: The Snow and Heat Transfer Metric (SHTM)**

A land model should be able to capture the exponential relationship between $A_{norm}$ and $S_{depth,eff}$, and it is useful to summarize the ability of a given model to do so as a compact metric. Here, we develop a Snow and Heat Transfer Metric (S$HTM$) that could be used within a broader land model analysis system such as the International Land Model Benchmarking project (ILAMB; http://www.ilamb.org; Luo et al., [2012]). The metric is designed to have a value from 0 to 1, and describes the

departure of a model's snow insulation curve from the observed curve. As the marginal influence of snow insulation decreases after a $S_{depth,eff}$ value of about 25cm and most of the seasonal snow regions have a $S_{depth,eff}$ below 30cm (Figure 5), the SHTM value is only calculated over the range of 0 to 30cm. For each 1cm of $S_{depth,eff}$, up to 30cm, twice the difference in $A_{norm}$ between the model and observational curves is obtained; a root mean square error of these values is then computed and subtracted from 1. The *SHTM* value is thus:

$$SHTM = 1. - \sqrt{\left[2*(Model\ A_{norm,\iota} - Observed\ A_{norm,\iota})\right]^2} \qquad (7)$$

The closer *SHTM* is to 1, the better the model is at reproducing the observed snow insulation curve; a lower limit of 0 is placed on the SHTM. An important feature of the S*HTM* is that it effectively isolates analysis of the snow insulation process and therefore should be robust across different climate forcing. We test that hypothesis here by calculating the *SHTM* for ten 15-year periods from 1850-2000 for each model. The mean 2m air temperature over the terrestrial Northern Hemisphere north of 55°N for the cooling period (Oct-Mar) was computed for each 15-year period for each model. As a measure of changing climates, the minimum and maximum of these 15-year averages was differenced per model, resulting in a span of 1.25°C to 4.15°C, with a mean of 2.15°C. Despite these climatic changes, the *SHTM* maintains a fairly constant value for each land model. Over the 150 years, all models return a *SHTM* standard deviation of less than 0.021, which is substantially less than the typical *SHTM* difference between models and indicates a general invariance to climate variability (Figure 6).

Due to scatter in the data (both modeled and observed data), the S*HTM* can only be used as a broader indicator of model skill. Based on the standard deviation of scores and observational uncertainty, *SHTM* values within 0.05 of each other may not be significantly different and incremental improvements to the snow insulation schemes may not necessarily result in meaningful changes in the *SHTM* score. Additionally, the *SHTM* does not evaluate the ability of models to simulate snow accumulation or ablation; additional datasets and metrics are needed to assess these processes.

**5 Conclusions**

Model evaluation is an important aspect of the overall modeling process: e.g., development, application and/or prediction. Land, hydrology and ecology models can range in complexity from simpler, more conceptually-based, formulations through to highly explicit representations that incorporate a multitude of processes. Regardless of their complexity, it is important that the underlying physical processes are represented correctly.

Deriving the relationship between a normalized temperature amplitude difference between air and soil, $A_{norm}$, and the seasonal effective snow depth, $S_{depth,eff}$, we have shown that observations are consistent with heat transfer theory. Analysis of observed results suggests that the marginal impact of snow insulation diminishes beyond a $S_{depth,eff}$ of about 25cm. Structural weaknesses in several models have been exposed by examining their ability to represent the atmosphere-land heat transfer process in the presence of snow. To quantitatively compare model performance, a snow and heat transfer metric, *SHTM*, was designed. The *SHTM* value per model changed little when calculated for different periods over 150 years of climate change, suggesting it can fairly unambiguously provide an indication of whether model structural/parameter deficiencies exist by negating other areas of uncertainty, such as the model forcing data. The *SHTM* is a useful model diagnostic that can be added to existing land model analysis / benchmarking systems.

## 6 Author Contribution

AGS and DML conceived the idea. AGS conducted the analysis. AGS, DML, and CDK wrote the manuscript.

## 7 Acknowledgements

Among the authors, this work was supported by NSF PLR-1304152, PLR-1304220, NASA IDS project 1517544, DOE DE-FC03-97ER62402/A010 and DE-AC02-05CH11231. Code for this method is to be implemented within the ILAMB system. Data sources and acknowledgements are given in the text.

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

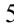

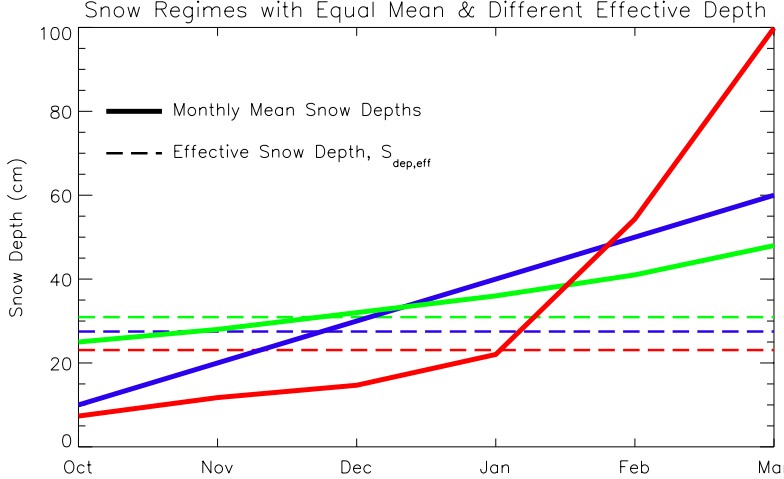

**Figure 1: Effective snow depth (S$_{depth,eff}$, dashed lines) of three different snow regimes that have equal mean values over the period October-March. Earlier snowfall receives greater weighting as it represents greater insulation; data in green thus has the greatest effective depth.**

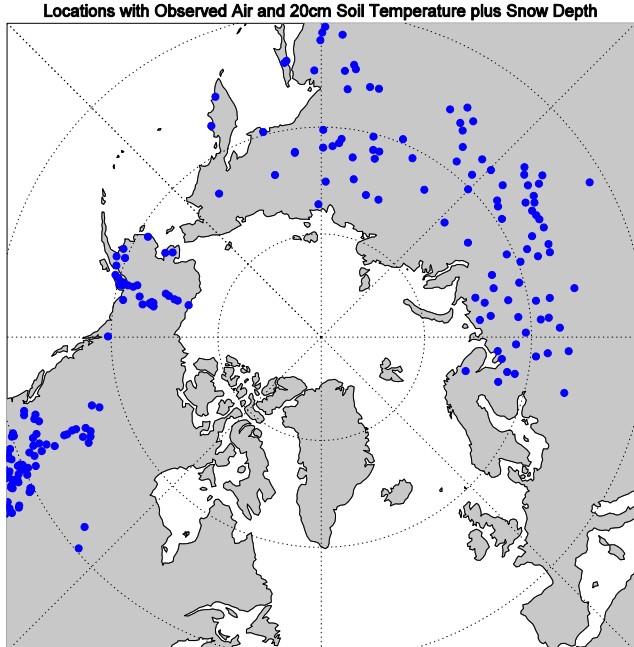

**Figure 2: Locations with co-located observations of air temperature, 20cm soil temperature, and snow depth.**

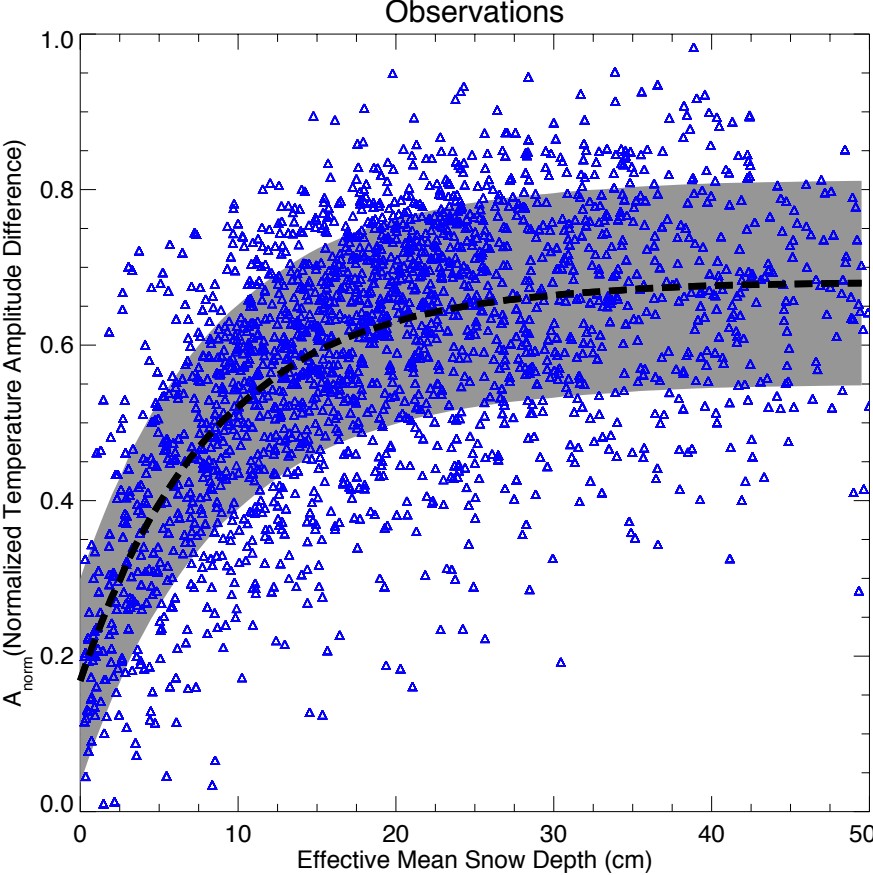

**Figure 3: Observed relation between $A_{norm}$ and effective mean snow depth ($S_{depth,\,eff}$) along with the resulting exponential fit (dashed line). The grey shading shows the median fit plus/minus the mean scatter ($O_{err}$) of all fits.**

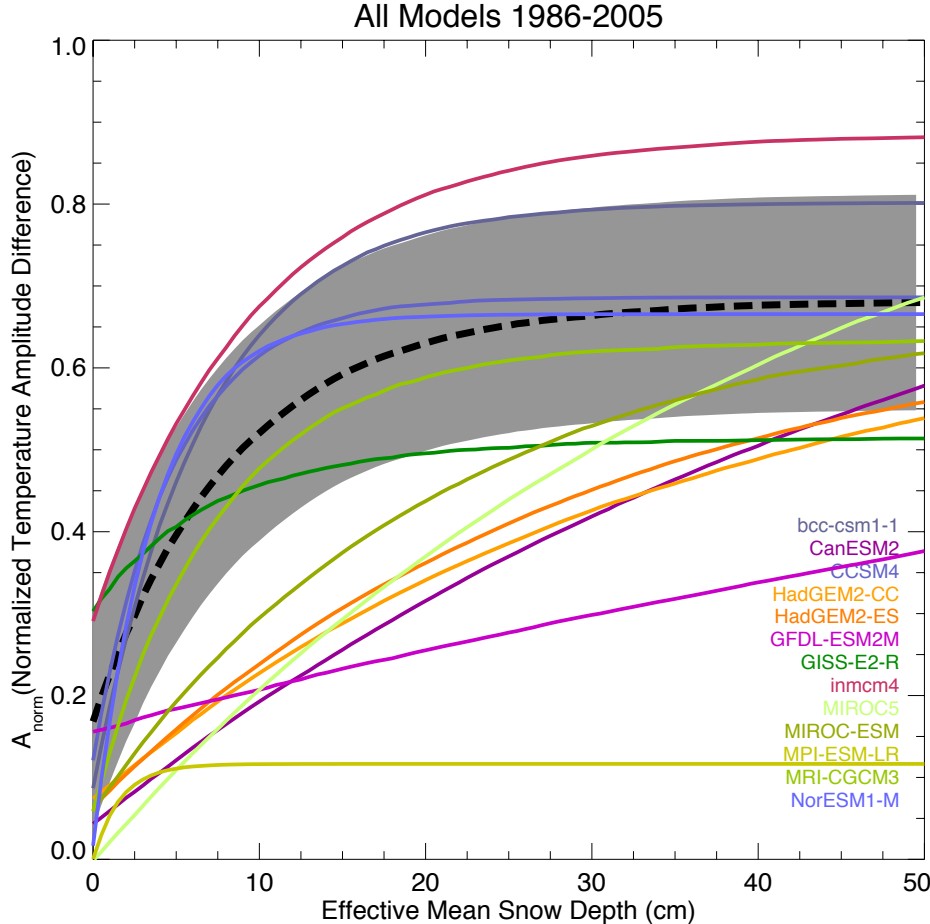

**Figure 4: Model fits to equation (5), showing large differences in how heat is transferred though the snowpack to the soil. Data from the CMIP5 comparison period 1986-2005 is used. The dashed black line is the observational fit with grey shading representing the observational scatter as in Figure 3.**

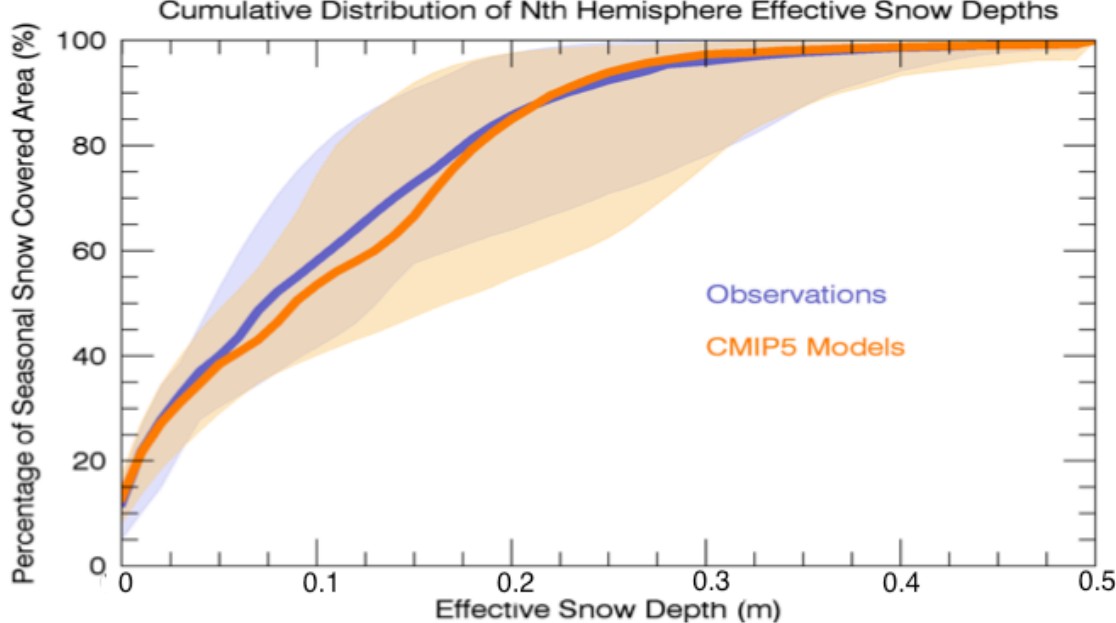

**Figure 5: Cumulative distribution curves for climatological effective snow depths ($S_{depth,eff}$) as a percentage of all seasonally snow covered area in the Northern Hemisphere. The observed and model climatologies span a minimum of 10 years in the last quarter of the 20$^{th}$ century. The bold lines show the median values of the curves for both observations and models; shading shows the total range of curves. Note that the absolute snow covered area can be quite different amongst models and observational estimates.**

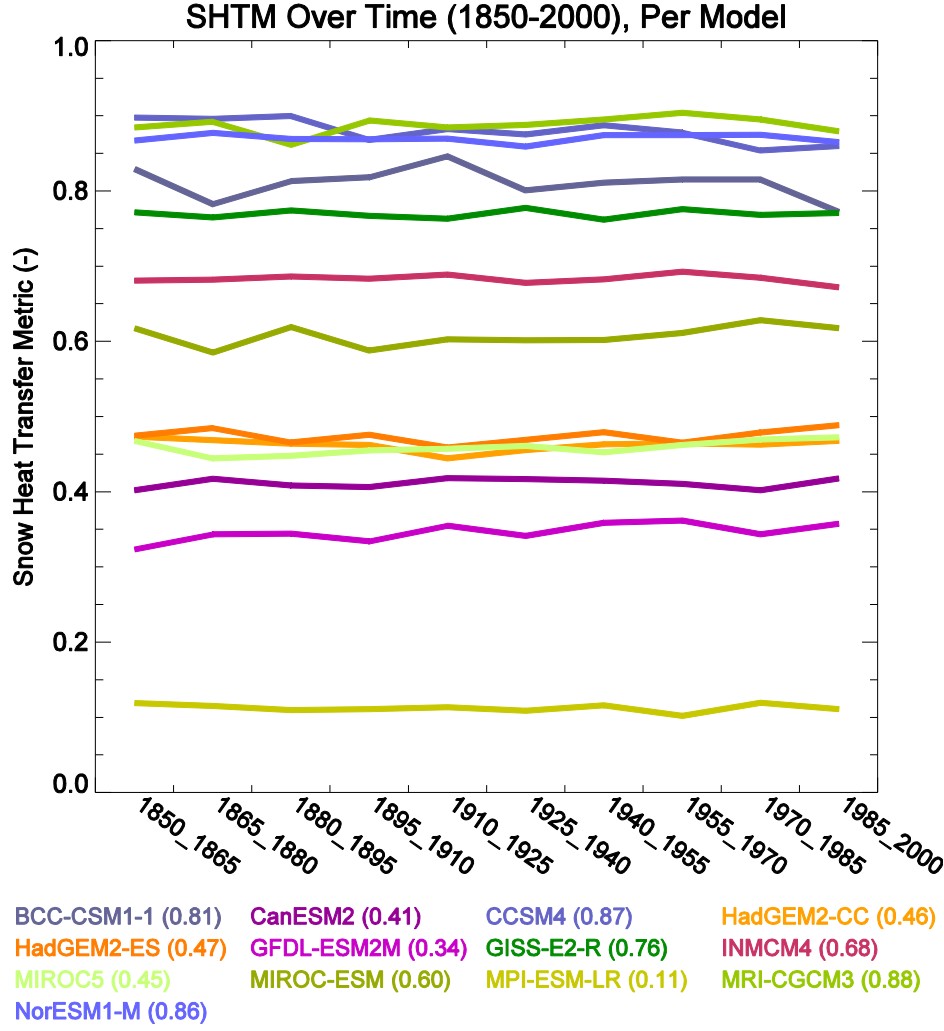

## SHTM Over Time (1850-2000), Per Model

BCC-CSM1-1 (0.81)   CanESM2 (0.41)   CCSM4 (0.87)   HadGEM2-CC (0.46)
HadGEM2-ES (0.47)   GFDL-ESM2M (0.34)   GISS-E2-R (0.76)   INMCM4 (0.68)
MIROC5 (0.45)   MIROC-ESM (0.60)   MPI-ESM-LR (0.11)   MRI-CGCM3 (0.88)
NorESM1-M (0.86)

**Figure 6: Mean value of the Snow and Heat Transfer Metric (SHTM) for each model. Values closer to 1.0 indicate better agreement with observations. SHTM values are relatively stable over 150 years of climate for each model.**