# Peer review of "Process-level model evaluation: A Snow and Heat Transfer Metric"

_The Cryosphere, 2016_

## Referee Comment (RC1) · Anonymous Referee #1 · 13 Dec 2016

This is an interesting and concise paper that proposes a compact method to evaluate the capacity of land surface models to represent the effect of snow inflation on the underlying soil. I have no doubt that the proposed metric, with some little changes proposed in the following, will be widely used. The paper is yet another illustration why the first author's recent passing away is a huge loss for the scientific community.

The figures are all relevant and easily readable. Relevant scientific literature is appropriately referenced. No unnecessary detail clutters the simple and clear message of the paper.

This work should therefore be published after a few minor changes suggested below.

Specific remarks.

[Figure]

- Page 2, line 12: The primary motivation is certainly a good representation of soil temperatures. One could add, however, that wrong temperatures at the snow/soil interface, caused by wrong snow conductivity, can feed back on the snow pack itself via a modified snow metamorphism (in cases models do simulate snow metamorphism dependent on temperature or vertical temperature gradients).

- Page 2, line 26: There is a little incoherence that could be acknowledged: The theory presented here initially supposes a periodic (sine) air temperature signal; however, the theory is then limited to the "cooling season".

- Page 3, line 3 : "2m air temperature serves as a sufficient proxy as the two quantities tend to equilibrate towards each other, particularly in colder months of high latitude regions with low solar input": Yes and no: In some cases (strong inversion), temperature difference between the snow-air interface and the air at 2 m height can be substantial.

- Equation 3: Why not use immediately $A_0$ and $A_z$ instead of introducing new variables $A_{air}$ and $A_{soil}$ which are not really used?

- Equation 6: The general form of this equation, in particular the numerator of the right hand side, makes sense, but the specific form of the denominator does not. The denominator (which is a constant) should be chosen such that if snow depth is constant (i.e. all snow falls in October), the efficient snow depth is equal to this constant value. Therefore the denominator should read: $\sum\limits_{n=1}^{M} n$ (or $(M+1)*M/2$, which is equivalent). For the case of the blue curve in figure 1, which is apparently $S(i) = i*0.1$ (with i=1 for October and i=6 for March), this would yield $S_{eff}=0.266$ m, which is less than the average depth of 0.35 m. This would make sense; in Figure 1, for the same case, $S_{eff}$ is higher than the simple time average, which is incoherent. By the way, I have the impression that equation 6 is not what is plotted in Figure 1. In any case, the difference is only a constant factor, so this has no important effect on the results presented in the rest of the paper. But I think that the definition of $S_{eff}$ should make immediate sense for simple cases. Right now, it does not.

- Equation 6: What would the results look like if the time period considered would be limited to the period before substantial snow melt occurs? In southerly areas, uno can already melt in March. Does this introduce noise?

- Page 5, line 12: Would it make sense, and would it change the results, to offset the snow depths by adding a positive constant corresponding to a slab of snow with equivalent thermal insulation as 20 cm of soil?

- Page 6, line 9: Some models have a vertical 'soil' axis that comprises the snow. That is, 'soil' depth is not counted from the soil-snow interface downwards, but it starts at the snow-atmosphere interface. That could explain some very far off outliers.

- Page 6, line 20: Yes, but the initial argumentation says that the metric presented here is valid in the case when there are no phase changes. (But the argument is correct nevertheless)

- Equation 7 is not particularly elegant. It must be artificially limited to exclude values below 0. A more elegant definition could be: $SHTM= SUM(MIN(A_{norm,obs,i},A_{norm,mod,i}))/SUM(MAX(A_{norm,obs,i},A_{norm,mod,i}))$ This would automatically yield values between 0 and 1 because $A_{norm}$ is always $>=0$. Other rather natural and coherent forms for the RHS of equation 7 can be easily defined.

---

## Referee Comment (RC2) · Anonymous Referee #2 · 23 Dec 2016

General comments: In this paper the authors develop a relatively straightforward diagnostic metric (SHTM - Snow Heat Transfer Metric) for establishing whether the heat transfer through the soil-snow layer is realistically simulated by a climate model. The diagnostic is based on the amplitude equation for the conductive heat flow which is integrated over the period when air temperature are below freezing to obtain the difference in the seasonal temperature amplitudes at some depth in the soil, and the effective snow depth which describes the insulating effect the snow layer over the accumulation season. The authors use observed air temperature, snow depth and soil temperatures at 20 cm from climate stations in Russia, Canada and the USA to obtain an estimate of the curve relating the effective snow depth to the normalized difference in temperature amplitude (Figure 3). There is considerable scatter around this curve which the authors describe as "noise" or "error". However, I suspect that the results shown in Figure 3

represent a number of different curves that reflect different snow-climate regions (e.g. Sturm et al. 1995) and soil properties (e.g. organic soils). The authors then compare the ability of 13 CMIP5 climate models to replicate the observed relationship derived from the surface observations using all land grid points north of 55 deg N (Figure 4). The results show major differences between models but one question that crops up at this point is whether the somewhat limited spatial sample of observations (Fig. 2) influences this comparison. Repeating the analysis for grid points nearest the observations would answer this question.

The large difference in SHTM between models is worrisome but we don't get any sense from the paper of the climatic consequences of a poor fit to the observed heat transfer relationship and how much of the poor fit is coming from representation of snowpack versus the specification of soil thermal properties. Presumably one would not use a model with a poor SHTM metric for studies of the soil thermal regime or permafrost, but apart from that I'm not quite sure what the metric tell us. The metric would certainly have value in evaluating the performance of different versions of climate models and land surface schemes. One aspect of the paper that could be expanded on (topic for follow-on paper?) is the spatial variability in SHTM in observations and models.

In conclusion this paper is a useful addition to the literature and a testament to Drew's ability to derive practical applications from complex processes.

Detailed comments:

- Page 1, line 30: Mudryk et al (2016) would be a useful reference to cite in this context as it specifically addresses the uncertainty issue in observational SWE datasets

Mudryk, L.R., C. Derksen, P.J. Kushner and R. Brown, 2015. Characterization of Northern Hemisphere snow water equivalent datasets, 1981–2010. Journal of Climate, 28:8037-8051.

- Page 5: Observed data. The authors have a rather limited sample for characterizing

the NH land area average amplitude used in eqn. (7). It would be instructive to provide the readers with some idea of the variability in Fig. 3 for a sample of the major snow-climate (e.g. Sturm et al. 1995) and ecoclimatic regions.

- Figure 4: I suggest you use "scatter" rather than the statistical term "error"

- Figure 5: the derivation of Figure 5 is not provided in the paper and there is no discussion of this Figure. This shows the CMIP5 ensemble close to the observations but this is a potentially misleading message.

- Figure 6: What about spatial variability in SHTM? Is this important? How does this vary between models? To what extent do the different geophysical fields used in models contribute to this variability i.e. how much of a model's behaviour in SHTM is related to representation of the snowpack versus specification of soil thermal properties?

---

## Referee Comment (RC3) · Anonymous Referee #3 · 27 Dec 2016

This is an interesting paper with a data-driven and process-focussed approach that typifies the recently deceased first author's work. I expect that it will prove to be useful for model evaluation; indeed, this method has already been suggested for inclusion in the methods for evaluating models in the upcoming Earth System Model – Snow Model Intercomparison Project (http://www.climate-cryosphere.org/activities/targeted/esm-snowmip).

The observed relations ship between "temperature amplitude difference" and "effective snow depth" shown in Figure 3 has a great deal of scatter. A lot of this scatter will come from genuine physical processes. It would be useful to have some discussion of the influences of soil texture, soil moisture and freezing on the results. Without separating out these influences, it doesn't appear that this method could provide very strong constraints on models, but it is likely to still be useful because current models,

as shown in Figure 4, have an even larger range. It would be interesting to know if the results of this paper can be related to the performances of the same models in simulating permafrost extent, as discussed by Koven et al. (2013). The Hadley Centre model is identified as one in which soil temperatures under snow track air temperatures too closely because of the simplicity of the snow model used. The developers of this model are well aware of this limitation and have implemented a multi-layer snow model to address it; the model is described by Best et al. (2011) and its impacts on permafrost simulations by Chadburn et al. (2015).

The definition of effective snow depth in Equation (6) is curious and requires explanation. Why is it chosen so as to give an effective depth that is greater than the average depth for any month for the green line in Figure 1?

page 2, line 31 "the period over which the forcing is applied" is ambiguous. Something like "the frequency of the forcing" would be better.

page 4, line 3 The R parameter is an effective damping depth, not an effective thermal diffusivity.

Best, MJ, and 16 others, 2011. The Joint UK Land Environment Simulator (JULES), model description. Part 1: Energy and water fluxes. Geoscientific Model Development, 4, 677–699.

Chadburn, SE, EJ Burke, RLH Essery, J Boike, M Langer, M Heikenfeld, PM Cox and P Friedlingstein, 2015. Impact of model developments on present and future simulations of permafrost in a global land-surface model. The Cryosphere, 9, 1505–1521.

---

## Author Comment (AC1) · 15 Mar 2017

**Reviewer comments in bold, responses in plain text.**

**Reviewer 1 comments**

**This is an interesting and concise paper that proposes a compact method to evaluate the capacity of land surface models to represent the effect of snow inflation on the underlying soil. I have no doubt that the proposed metric, with some little changes proposed in the following, will be widely used. The paper is yet another illustration why the first author's recent passing away is a huge loss for the scientific community. The figures are all relevant and easily readable. Relevant scientific literature is appropriately referenced. No unnecessary detail clutters the simple and clear message of the paper.**

We thank the reviewer for his comments. We agree that the simplicity of the metric is one of its strengths. We also appreciate the comments about Drew's passing and how his passing is a huge loss for the scientific community. We couldn't agree more.

**This work should therefore be published after a few minor changes suggested below.**

**Specific remarks.**

**- Page 2, line 12: The primary motivation is certainly a good representation of soil temperatures. One could add, however, that wrong temperatures at the snow/soil interface, caused by wrong snow conductivity, can feed back on the snow pack itself via a modified snow metamorphism (in cases models do simulate snow metamorphism dependent on temperature or vertical temperature gradients).**

This is a good point. We have added the following sentence: "Additionally, biases in the simulated temperature at the snow/soil interface can adversely affect the snow pack itself though the impact of these biases on snow metamorphism at the base of the snow pack."

**- Page 2, line 26: There is a little incoherence that could be acknowledged: The theory presented here initially supposes a periodic (sine) air temperature signal; however, the theory is then limited to the "cooling season".**

This is correct. We believe that the text is already relatively clear on this. E.g., we note: "The above theory is adapted to the cooling period of the year, defined here as October to March."

**- Page 3, line 3 : "2m air temperature serves as a sufficient proxy as the two quantities tend to equilibrate towards each other, particularly in colder months of high latitude regions with low solar input": Yes and no: In some cases (strong inversion), temperature difference between the snow-air interface and the air at 2 m height can be substantial.**

True. There is some error associated with differences (positive or negative) between air temperature at 2m and the temperature at the snow-air interface. We believe that the errors associated with this discrepancy have less of an impact on th metric than errors in the measurements themselves. But, we do now include this statement to acknowledge this point. "The actual land surface temperature is rarely observed *in situ*, but 2m air temperature serves as a sufficient proxy as the two quantities tend to equilibrate towards each other, particularly in colder months of high latitude regions with low solar input, though in situations with strong inversions, the temperature difference between the snow-air interface and 2m height can be substantial (this is an acknowledged, yet unavoidable limitation)."

**- Equation 3: Why not use immediately $A_0$ and $A_z$ instead of introducing new variables $A_{air}$ and $A_{soil}$ which are not really used?**

Certainly, one could use A_0 and A_z directly, but we feel that it is actually easier to understand what A_norm is the way it is presented so we have elected to maintain as in the original document.

**- Equation 6: The general form of this equation, in particular the numerator of the right hand side, makes sense, but the specific form of the denominator does not. The denominator (which is a constant) should be chosen such that if snow depth is constant (i.e. all snow falls in October), the efficient snow depth is equal to this constant value. Therefore the denominator should read: $\sum\limits_{n=1}^{M} n$ (or $(M+1)*M/2$, which is equivalent). For the case of the blue curve in figure 1, which is apparently $S(i) = i*0.1$ (with i=1 for October and i=6 for March), this would yield S_eff=0.266 m, which is less than the average depth of 0.35 m. This would make sense; in Figure 1, for the same case, S_eff is higher than the simple time average, which is incoherent. By the way, I have the impression that equation 6 is not what is plotted in Figure 1. In any case, the difference is only a constant factor, so this has no important effect on the results presented in the rest of the paper. But I think that the definition of S_eff should make immediate sense for simple cases. Right now, it does not.**

Agreed.  We have fixed the equation and replotted Figure 1, Figure 3, Figure 4, and Figure 5

**- Equation 6: What would the results look like if the time period considered would be limited to the period before substantial snow melt occurs? In southerly areas, it can already melt in March. Does this introduce noise?**

We tested with various limits to cooling season and the results are qualitatively similar.

**- Page 5, line 12: Would it make sense, and would it change the results, to offset the snow depths by adding a positive constant corresponding to a slab of snow with equivalent thermal insulation as 20 cm of soil?**

We do not think that this would add any value.  There is an offset in the thermal insulation at 0 effective snow depth that represents the thermal offset between air temperatures and 20cm soil temperature.  We feel that the way the results are presented now make this clear and that doing something like 'replacing' the snow with a slab of snow would reduce clarity.

**- Page 6, line 9: Some models have a vertical 'soil' axis that comprises the snow. That is, 'soil' depth is not counted from the soil-snow interface  downwards, but it starts at the snow-atmosphere interface. That could explain some very far off outliers.**

We agree and we have already noted this in the discussion.  "Incorrect snow heat transfer curves are symptomatic of model deficiencies. As an example, the land scheme in the Hadley Center models used here [MOSES2.2; Essery et al., 2001] applies a composite snow model where the top soil layer and snowpack share the same temperature [Slater et al., 2001], hence insulation is not properly accounted for and cold temperatures easily penetrate into the soil."

**- Page 6, line 20: Yes, but the initial argumentation says that the metric presented here is valid in the case when there are no phase changes. (But the argument is correct nevertheless)**

That's correct.  We have elected to remove this statement because it is not a deficiency that is relevant to the snow insulation and therefore is outside the scope of this paper and has been noted previously in other studies.

**- Equation 7 is not particularly elegant. It must be artificially limited to exclude values below 0. A more elegant definition could be: SHTM= SUM(MIN(A_{norm,obs,i},A_{norm,mod,i}))/SUM(MAX(A_{norm,obs,i},A_{norm,mod,i})) This would automatically yield values between 0 and 1 because A_norm is always >=0. Other rather natural and coherent forms for the RHS of equation 7 can be easily defined.**

True.  Alternative forms of this equation could be implemented, but this is the form that Dr. Slater implemented and we prefer to leave this as is.   Alternative forms of the equation, while potentially more elegant, will not yield anything different in terms of results.

---

## Author Comment (AC2) · 15 Mar 2017

**Reviewer comments in bold, responses in plain text.**

**General comments: In this paper the authors develop a relatively straightforward diagnostic metric (SHTM - Snow Heat Transfer Metric) for establishing whether the heat transfer through the soil-snow layer is realistically simulated by a climate model. The diagnostic is based on the amplitude equation for the conductive heat flow which is integrated over the period when air temperature are below freezing to obtain the difference in the seasonal temperature amplitudes at some depth in the soil, and the effective snow depth which describes the insulating effect the snow layer over the accumulation season. The authors use observed air temperature, snow depth and soil temperatures at 20 cm from climate stations in Russia, Canada and the USA to obtain an estimate of the curve relating the effective snow depth to the normalized difference in temperature amplitude (Figure 3). There is considerable scatter around this curve which the authors describe as "noise" or "error". However, I suspect that the results shown in Figure 3 represent a number of different curves that reflect different snow-climate regions (e.g. Sturm et al. 1995) and soil properties (e.g. organic soils).**

We agree and we have updated the text to make it clear that by noise / error, we are referring to observational error as well as the range of curves that arise due to different snow / soil regimes. We have rewritten to "The observations show the expected exponential shape and fit the underlying theory (5) well despite significant scatter in the data (Figure 3). The scatter is likely due to several things including measurement error, the range of conditions and snow regimes [Sturm et al. 1995], that occur across the landscape, including the timing of snowfall, the pattern of snow metamorphism, the properties and moisture content of the soil as well as uncertainty about the measurement locations of the observed data."

**The authors then compare the ability of 13 CMIP5 climate models to replicate the observed relationship derived from the surface observations using all land grid points north of 55 deg N (Figure 4). The results show major differences between models but one question that crops up at this point is whether the somewhat limited spatial sample of observations (Fig. 2) influences this comparison. Repeating the analysis for grid points nearest the observations would answer this question. The large difference in SHTM between models is worrisome but we don't get any sense from the paper of the climatic consequences of a poor fit to the observed heat transfer relationship and how much of the poor fit is coming from representation of snowpack versus the specification of soil thermal properties.**

The curves from the models are robust whether or not we sample at the same locations as the obs or globally, though obviously there is more scatter when sampling fewer points.

It is beyond the scope of this study to assess the climatic consequences of a poor representation of snow heat transfer. The one thing that is clear is that permafrost simulations will suffer if snow heat transfer is not represented accurately. To determine broader climatic consequences would require additional climate model sensitivity runs with a range of snow heat transfer representations / parameters.

Because the observations of soil temperature are not at the surface, it isn't possible to fully distinguish where in the snow/upper soil system a disagreement between models and observations might arise. However, the offset in Anorm at zero effective snow depth is an indication of the impact of soil heat transfer between the soil surface and 20cm depth. In obs, this value ranges from about 0.05 to 0.3. Many models exhibit a lower normalized temperature difference value at zero effective snow depth compared to obs, which means that they transfer heat through the soil too efficiently. The sources of a soil heat transfer bias are myriad and could be due to biases/errors in soil texture including organic matter, soil moisture and soil moisture phase, and soil thickness. Many to most of these models do not represent soil organic matter (which is highly insulative), so this is a potential source of bias in these models. We have added a paragraph to discuss this point.

**Presumably one would not use a model with a poor SHTM metric for studies of the soil thermal regime or permafrost, but apart from that I'm not quite sure what the metric tell us. The metric would certainly have value in evaluating the performance of different versions of climate models and land surface schemes. One aspect of the paper that could be expanded on (topic for follow-on paper?) is the spatial variability in SHTM in observations and models. In conclusion this paper is a useful addition to the literature and a testament to Drew's ability to derive practical applications from complex processes.**

The metric tells us that models with a poor SHTM metric are not correctly modeling the thermal insulation of snow and it should be used to assess the quality of different models and potentially identify what models are fit for purpose (e.g., as reviewer notes, a model with poor SHTM should likely not be utilized in permafrost studies). See below for more comments on spatial variability of SHTM, but basically we are not convinced that it is appropriate, without a lot more observations, to study the spatial variability of SHTM. Our goal here is to generate a constraint on the representation of snow heat transfer in models that can be applied globally.

**Detailed comments:**
**- Page 1, line 30: Mudryk et al (2016) would be a useful reference to cite in this context as it specifically addresses the uncertainty issue in observational SWE datasets**
**Mudryk, L.R., C. Derksen, P.J. Kushner and R. Brown, 2015. Characterization of Northern Hemisphere snow water equivalent datasets, 1981–2010. Journal of Climate, 28:8037-8051.**

Good suggestion. We have added the reference.

**- Page 5: Observed data. The authors have a rather limited sample for characterizing the NH land area average amplitude used in eqn. (7). It would be instructive to provide the readers with some idea of the variability in Fig. 3 for a sample of the major snow climate (e.g. Sturm et al. 1995) and ecoclimatic regions.**

It would certainly be good to be able to see how the curve differs for different snow climate regimes, but we are limited by the availability of collocated snow, air, and soil temperature data. As noted in the text, at least some of the significant scatter likely arises from the different snow climates that are sampled which lead to different snowpack densities across ecoclimatic regions. The best we can do is to note that some of the scatter is likely attributable to these factors and to further note that snowpacks with seasonal snowpack dynamics and average densities that lie outside our sampling could generate different curves.

**- Figure 4: I suggest you use "scatter" rather than the statistical term "error"**

Good point. We have modified to using the term 'scatter' rather than 'error' throughout the paper.

**- Figure 5: the derivation of Figure 5 is not provided in the paper and there is no discussion of this Figure. This shows the CMIP5 ensemble close to the observations but this is a potentially misleading message.**

We aren't clear what the reviewer thinks is missing. The derivation of the figure is in the figure caption and is pretty straightforward. It's true that the CMIP5 ensemble resembles the observations in terms of this diagnostic, which is only indicating that shallow snow depths are more common in both observations and models than deep ones. This seems uncontroversial. Not sure what additional discussion would be helpful.

**- Figure 6: What about spatial variability in SHTM? Is this important? How does this vary between models? To what extent do the different geophysical fields used in models contribute to this variability i.e. how much of a model's behaviour in SHTM is related to representation of**

**the snowpack versus specification of soil thermal properties?**

There may be some spatial variability in the SHTM, but that is not really the point. In Figure 6, we are emphasizing that the metric is relatively insensitive to climate forcing since the values remain constant through time and with climate change. One needs quite a bit of data to create the functional relationship curves so at best one could potentially create a map of very large regions of SHTM scores, but since the underlying data generating the observed curve is quite sparse, it doesn't really make sense to make a map of SHTM.

It is not possible to identify from these standard CMIP5 model runs where the source of discrepancy between model behavior and the obs comes from. That said, one can infer that the offset of approximately 0.05-0.3 in the observed normalized temperature difference in Figure 4 at zero effective snow depth, reflects the impact of the soil. Models that have a low normalized temperature difference value at zero effective snow depth compared to obs likely transfer heat through the soil too efficiently. The sources of a soil heat transfer bias are myriad and could be due to biases/errors in soil texture including organic matter, soil moisture and soil moisture phase, and soil thickness.

---

## Author Comment (AC3) · 15 Mar 2017

**Reviewer comments in bold, responses in plain text.**

**This is an interesting paper with a data-driven and process-focussed approach that typifies the recently deceased first author's work. I expect that it will prove to be useful for model evaluation; indeed, this method has already been suggested for inclusion in the methods for evaluating models in the upcoming Earth System Model – Snow Model Intercomparison Project (http://www.climatecryosphere.org/activities/targeted/esm-snowmip). The observed relations ship between "temperature amplitude difference" and "effective snow depth" shown in Figure 3 has a great deal of scatter. A lot of this scatter will come from genuine physical processes. It would be useful to have some discussion of the influences of soil texture, soil moisture and freezing on the results. Without separating out these influences, it doesn't appear that this method could provide very strong constraints on models, but it is likely to still be useful because current models, as shown in Figure 4, have an even larger range.**

We agree that the scatter in the observations can arise from multiple sources and in the revised version of the paper we better acknowledge this at the start of the Results and Discussion section: "The observations show the expected exponential shape and fit the underlying theory (5) well despite significant scatter in the data (Figure 3). The scatter likely arises from several sources including (1) the range of climate conditions and snow regimes [Sturm et al. 1995] that occur across the landscape, including the timing of snowfall and the pattern of snow metamorphism, (2) the properties and moisture content of the soil, and (3) uncertainties in the measurements themselves and the measurement locations of the observed data.  It is not possible to distinguish which of these sources of uncertainty dominates the scatter seen in Figure 3."

Additionally, we note that we have already included discussion of the limitations of the SHTM metric and noted that small differences in a score are not necessarily indicative of a significant improvement.  And, as the reviewer notes, current generation models show a wide range of performance for this metric.  Our perspective is that even a weak constraint is better than no constraint and that if all models could be updated so that they lie somewhere near or within the gray shading in Figures 3/4, that would constitute a big improvement from an Earth System modeling perspective.

**It would be interesting to know if the results of this paper can be related to the performances of the same models in simulating permafrost extent, as discussed by Koven et al. (2013). The Hadley Centre model is identified as one in which soil temperatures under snow track air temperatures too closely because of the simplicity of the snow model used. The developers of this model are well aware of this limitation and have implemented a multi-layer snow model to address it; the model is described by Best et al. (2011) and its impacts on permafrost simulations by Chadburn et al. (2015).**

We have added the following statement to acknowledge that this issue has been resolved in the Hadley Center model. "Note that the Hadley Centre model developers have addressed this limitation by implementing a multi-layer snow model [Best et al., 2011; Chadburn et al. 2015]."

**The definition of effective snow depth in Equation (6) is curious and requires explanation. Why is it chosen so as to give an effective depth that is greater than the average depth for any month for the green line in Figure 1?**

We have updated the equation as per the suggestion of reviewer 1 and redrawn the figure.

**page 2, line 31 "the period over which the forcing is applied" is ambiguous. Something like "the frequency of the forcing" would be better.**

To improve clarity, we  modify the sentence to "The value of d is a function of the thermal diffusivity

of the medium and the length of time that the forcing is applied."

**page 4, line 3 The R parameter is an effective damping depth, not an effective thermal diffusivity.**

Corrected.

**Best, MJ, and 16 others, 2011. The Joint UK Land Environment Simulator (JULES), model description. Part 1: Energy and water fluxes. Geoscientific Model Development, 4, 677–699.**
**Chadburn, SE, EJ Burke, RLH Essery, J Boike, M Langer, M Heikenfeld, PM Cox and P Friedlingstein, 2015. Impact of model developments on present and future simulations of permafrost in a global land-surface model. The Cryosphere, 9, 1505–1521.**